# Accounting for isoform expression increases power to identify genetic regulation of gene expression

**Nathan LaPierre**[1,2]*, **Harold Pimentel**[3,4,5]*

**1** Department of Computer Science, University of California, Los Angeles, California, United States of America, **2** Department of Human Genetics, University of Chicago, Illinois, United States of America, **3** Department of Human Genetics, University of California, Los Angeles, California, United States of America, **4** Howard Hughes Medical Institute, Chevy Chase, Maryland, United States of America, **5** Department of Computational Medicine, University of California, Los Angeles, California, United States of America

* nlapier2@uchicago.edu (NLP); hjp@g.ucla.edu (HP)

**Data Availability Statement:** The code used for simulations, preprocessing, running methods, and evaluating results can be found at https://github.com/nlapier2/isoQTL. All data used in this manuscript is publicly available with the exception

## Abstract

A core problem in genetics is molecular quantitative trait locus (QTL) mapping, in which genetic variants associated with changes in the molecular phenotypes are identified. One of the most-studied molecular QTL mapping problems is expression QTL (eQTL) mapping, in which the molecular phenotype is gene expression. It is common in eQTL mapping to compute gene expression by aggregating the expression levels of individual isoforms from the same gene and then performing linear regression between SNPs and this aggregated gene expression level. However, SNPs may regulate isoforms from the same gene in different directions due to alternative splicing, or only regulate the expression level of one isoform, causing this approach to lose power. Here, we examine a broader question: which genes have at least one isoform whose expression level is regulated by genetic variants? In this study, we propose and evaluate several approaches to answering this question, demonstrating that "isoform-aware" methods—those that account for the expression levels of individual isoforms—have substantially greater power to answer this question than standard "gene-level" eQTL mapping methods. We identify settings in which different approaches yield an inflated number of false discoveries or lose power. In particular, we show that calling an eGene if there is a significant association between a SNP and any isoform fails to control False Discovery Rate, even when applying standard False Discovery Rate correction. We show that similar trends are observed in real data from the GEUVADIS and GTEx studies, suggesting the possibility that similar effects are present in these consortia.

## Introduction

Genome-Wide Association Studies (GWAS) have identified numerous genetic loci that are associated with a variety of traits and diseases [1, 2]. A core challenge is understanding the functional role of variants within these loci. Many loci identified in GWAS lie in non-coding regions of the genome, so their functional role is not immediately clear, though it is expected that many such loci play a role in gene regulation [3]. Molecular Quantitative Trait Loci (QTL)

of the GTEx genotypes, which must be obtained by request as described in https://gtexportal.org/home/protectedDataAccess. The GTEx expression data, covariates, and population information can be obtained from https://gtexportal.org/home/datasets. The GEUVADIS genotype data can be obtained online from https://www.ebi.ac.uk/arrayexpress/experiments/E-GEUV-1/files/ and the expression data can be obtained online from https://www.internationalgenome.org/data-portal/data-collection/geuvadis. Gene annotations were obtained via the Ensembl GRCh38 build 88 GTF file, which can be obtained from http://ftp.ensembl.org/pub/release-88/gtf/homo_sapiens/.

**Funding:** This work was partially funded by HHMI Hanna H Gray and Sloan fellows programs to HP. NL was funded in part by NSF award 2106908 and NIH awards U01HG011715 and R56HG010812. The funders had no role in study design, data collection and analysis, decision to publish, or preparation of the manuscript.

**Competing interests:** The authors declare no competing interests.

mapping studies help address this challenge. Molecular QTL studies identify genetic variants (often Single Nucleotide Polymorphisms, or "SNPs") that are associated with the molecular phenotypes under study. In this manuscript, we focus on expression QTL (eQTL) mapping, in which the molecular phenotype is gene expression, and a gene is called an "eGene" when at least one nearby SNP is significantly associated with its expression level. Large-scale eQTL mapping studies have led to a tremendous amount of information about which genetic variants regulate gene expression [4–7].

For any given DNA sequence comprising a "gene", different combinations of exons can be transcribed into mature mRNA (referred to as "alternative splicing"); the resulting distinct canonical mRNA sequences for a given gene are known as "isoforms" of the gene. Transcripts (copies) of these different isoforms are the actual biomolecules that are sequenced, such that each gene corresponds to several distinct mRNA sequences with different abundances. Several of the most-used eQTL mapping methods, such as Matrix eQTL [8] and FastQTL [9] do not explicitly provide a way to account for the abundances of the different isoforms. Accordingly, an important question is how to deal with these different isoform abundances when one's goal is to identify eGenes. It is common when using these tools to either quantify expression level at the gene level, or to quantify expression levels of individual isoforms and then sum or average these quantities before performing eQTL analysis [10, 11]. Large-scale studies have identified many eQTLs for a variety of eGenes using this strategy [4–7].

Underlying this approach is the assumption that all isoforms follow similar gene-level effects. However, this may not always be the case; isoforms can be functionally different at either the regulatory or protein region [12–14] and thus may be regulated differently. This approach loses power to identify eGenes if isoform expression levels are regulated in opposite directions by a SNP (e.g. splicing QTLs) or if a SNP only affects an isoform with a low abundance relative to the gene's other isoforms [11]. While several approaches accounting for isoform expression have been developed in the context of differential expression testing [11, 15, 16], these methods are designed for small samples and thus lack power for large-scale eQTL studies. For example, it is common to moderate the estimated variance of gene expression, thus resulting in a biased, albeit more stable, estimate of variance. Recently, QTLtools [17] extended the FastQTL method [9] to handle "grouped" hypotheses, i.e. testing effects on isoform-level expression for multiple isoforms in the same gene "group", but we are not aware of any extensive validation of these approaches beyond a few comparisons between different grouped tests in the QTLtools paper.

In this study, we propose and examine the performance of several approaches for accounting for isoform expression in eQTL mapping, which we collectively refer to as "isoform-aware methods". Specifically, we focus on the task of identifying eGenes, which we define here as genes for which one or more *cis*-SNPs are associated with the expression levels of one or more of the gene's isoforms. The notion of "eQTLs" and "eGenes" in this manuscript is therefore somewhat broader than the common definition that eQTLs are *cis*-SNPs associated with the "gene-level" expression of (e)Genes—i.e. the summed isoform expression levels. It captures "regulation of expression" in a broader sense, including events that affect specific isoforms of genes, such as alternative splicing or polyadenylation. For the remainder of the manuscript, we use the terms "eQTLs" and "eGenes" in this broader sense for the sake of simplicity, though alternative terms such as "iso-eQTLs" and "iso-eGenes" could be used.

We examine four types of approaches in this manuscript: (i) traditional p-value aggregation methods; (ii) "grouped" permutation tests implemented in the QTLtools package; (iii) general linear models; and (iv) multiple regression. The first three categories of methods use information obtained by regressing *cis*-SNPs against the expression levels of each isoform for a gene. P-value aggregation methods then attempt to aggregate the p-values from individual

regressions into a single p-value at the gene level. QTLtools can obtain a gene-level p-value from isoform-level p-value by employing a permutation scheme that generates a null distribution over all isoforms for a gene. Taking the results of the individual regressions as a "stacked" general linear model, one can perform a multivariate hypothesis test on whether a SNP has a nonzero association with any isoform. On the other hand, multiple regression between each *cis*-SNP and all isoforms can be performed, followed by an F-test for whether the SNP is associated with any isoform. *To our knowledge, the use of a general linear model for isoform-aware eQTL mapping is novel, and while p-value aggregation (such as Fisher's method* [18]*) and F-tests are standard analysis tools, their use in eQTL mapping is currently uncommon.* We explore these approaches in more detail below, compare their performance in simulations, and apply them to real expression data.

We demonstrate in simulations that isoform-aware methods offer a substantial increase in power to identify eGenes compared with the approach of summing or averaging isoform expression levels and then applying FastQTL or QTLtools (we refer to the latter approach as "QTLtools-mean" below). Corroborating these findings, we show that these methods find many more eGenes than the QTLtools-mean approach when run on isoform expression levels of 87 Yoruban individuals from the GEUVADIS study [4] and when applied to 494 European thyroid samples from the GTEx v8 dataset [6]. We identify several methods that have deflated power or inflated empirical False Discovery Rates (FDR) when non-*cis* effects are simulated, and demonstrate that a similar trend is observed in the GEUVADIS and GTEx data analyses. We also show that running QTLtools on individual isoform expression levels and then calling an eGene if there is a significant association between a SNP and any isoform fails to control FDR, even when applying FDR correction.

Finally, we compare and contrast the problem setup in this paper with approaches that find different types of QTLs such as splice QTLs or utilize allele-specific mapping information, and discuss other biological association testing problems where the approaches assessed in this paper should be useful. We expect that the findings discussed in this paper will serve as a useful guidepost for further isoform-aware approaches that aim to increase power to identify genetic regulation of gene expression.

## Results

### Overview of methods

We assessed the performance of several approaches towards utilizing isoform expression levels in eQTL/eGene mapping. Here the goal is to identify eGenes (and their associated "eSNPs"), which we define here as genes for which one or more *cis*-SNPs are associated with the expression levels of one or more of the gene's isoforms. For the purposes of this paper, we assume that the expression levels of each isoform for each individual have already been quantified by a method such as kallisto [23] or Salmon [24], and that only the association step is desired. Other use cases are discussed in the Discussion section. We describe the approaches below, and they are summarized in Table 1.

A common approach to this problem is to simply add the expression levels of the isoforms together to obtain expression levels for a gene, and then perform simple linear regression between each SNP in *cis* and the obtained gene expression levels. Alternatively, collapsing the isoform annotation to a "gene" annotation and "gene counting" is a possibility [25]. Methods such as Matrix eQTL [8] and QTLtools [17] can be applied to rapidly perform such association tests. Additionally, QTLtools introduces an approach to correct for the multiple hypothesis testing burden that occurs as a result of testing association with every SNP in *cis*, which are correlated due to linkage disequilibrium (LD). This approach uses a Beta distribution to

**Table 1. Summary of the methods evaluated in this manuscript.**

| Method Name | Description |
|---|---|
| QTLtools-sum/mean | Sum/average the expression quantities of all isoforms for a gene, then run simple linear regression with QTLtools [17]. |
| QTLtools-pca1 | Apply QTLtools' "pca1" method, taking the first principal component of the isoform expression levels and performing simple linear regression [17]. |
| QTLtools-best | Apply QTLtools' "best" method for applying a permutation test across all isoforms for a gene [17]. |
| Fisher/Cauchy/Min-perm | Perform simple linear regression against each isoform, then apply a p-value aggregation method, then perform permutation testing to correct for multiple hypotheses. The aggregation methods assessed were Fisher's method [18], the Cauchy Combination Test [19], or using the minimum p-value. |
| Fisher/Cauchy/Min-qtltools-iso | Run QTLtools [17] on individual isoforms, then apply a p-value aggregation method to those results. The aggregation methods assessed were Fisher's method [18], the Cauchy Combination Test [19], or using the minimum p-value. |
| F-test | Perform multiple regression between a SNP and all isoforms for a gene, then perform an F-test for significance. |
| Wilks-Bartlett | Perform simple linear regression against each isoform, then perform a multivariate hypothesis test for the effect of a SNP on any isoform [20], yielding a Wilks' Lambda Distribution [21] that Bartlett [22] showed is approximately chi-square distributed. |

approximate extreme p-values that would be obtained by an exhaustive permutation test [9, 17] (Methods). False Discovery Rate (FDR) control on the gene level can then be performed by applying a standard FDR control procedure such as the q-value of Storey and Tibshirani [26].

While this approach is effective at controlling FDR [9, 17], it can lose power when a SNP regulates isoforms in opposite directions (e.g. splice QTLs) or when one isoform is much more highly expressed than another and masks the effect of the SNP on a low-expressed isoform [11]. Several potential approaches mitigate these issues by taking into account the associations between the expression level of each isoform and the target SNP and then combine the association signal across all isoforms. We collectively refer to these approaches as "isoform-aware methods" (Fig 1). The first step in most of these approaches is to perform simple linear regression between a SNP and each isoform for a gene. The differences emerge in how to aggregate these association signals.

One such approach for isoform aggregation was proposed in the QTLtools paper [17]. Instead of computing permutation test p-values for each isoform separately, this approach involves permuting all isoforms for a gene at once. The individual SNP-isoform associations are then compared to this null distribution, and the pair with the best p-value is reported. Alternatively, the first principal component of the isoforms can be computed and then tested for association with each SNP. QTLtools also has an option to compute the mean of the isoforms and then test those phenotypes for association, but this is equivalent up to a constant term with summing the isoform expression levels. We refer to these approaches as "QTLtools-best", "QTLtools-pca1", "QTLtools-sum", and "QTLtools-mean" below.

Another approach is to apply p-value aggregation on the isoform regressions to produce a single gene-level p-value. Several common methods are simply taking the minimum p-value, using the classic Fisher's method [18], or using the popular recently-proposed Cauchy aggregation test, also known as "ACAT" [19, 27]. One relevant question that we investigated is whether it is appropriate to apply the p-value aggregation methods after the permutation-adjusted p-value is obtained via the method described in the FastQTL paper [9]—in which case one could simply run a method like QTLtools on individual isoforms, then use one of these methods to obtain gene-level p-values—or whether aggregation should be performed before permutation testing.

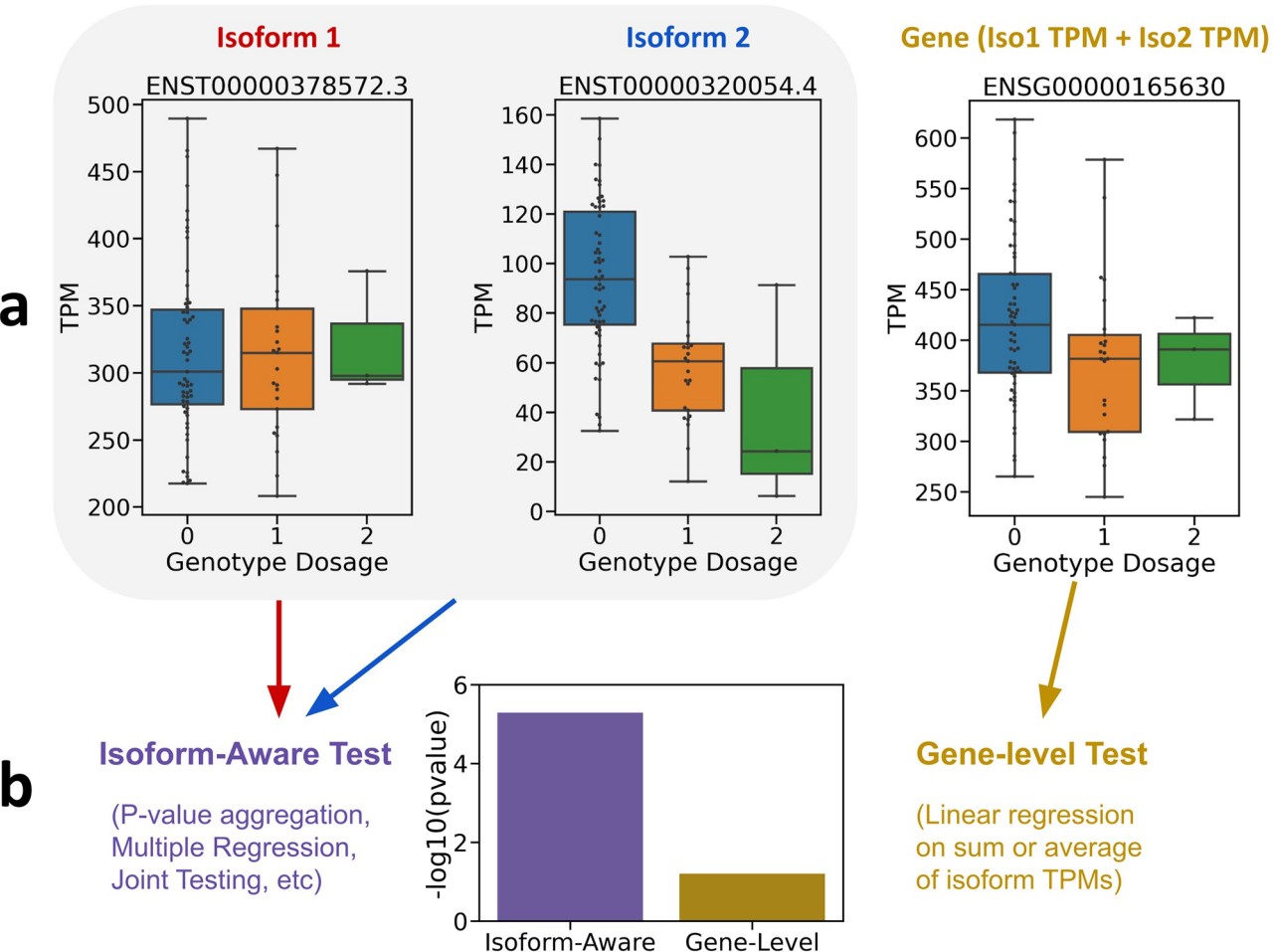

**Fig 1.** (a) Box plots depicting expression level measured in Transcripts Per Million (TPM; y-axis) versus the genotype of the most-associated SNP (x-axis) for the gene ENSG00000165630 and its two isoforms in the GEUVADIS [4] data. The gene plot shows the TPM obtained from summing the TPM of each constituent isoform. (b) The performance of two approaches to association is contrasted: "isoform-aware" tests which utilize association between the SNP and each isoform, versus "gene-level" tests which only test for association with the summed gene expression. The p-values depicted were obtained by running QTLtools [17] with its grouped permutation scheme (left) or on summed gene-level expression data (right).

A third approach is to apply a multivariate hypothesis test to the association statistics obtained from the regressions on each isoform to jointly assess whether the SNP in question is associated with any of the isoforms. Several formulations construing the problem as either a general linear model or as seemingly unrelated regressions (SUR) can be developed and are often used in multivariate analysis of variance (MANOVA) [20, 28]. Here, we focus on one of the best-studied approaches to this problem, which uses the Wilks' Lambda distribution [21]. Bartlett [22] developed a chi-squared approximation to the Wilks' Lambda, so that a chi-squared test can be applied to obtain a p-value for association between the SNP and any isoform [20]. One advantage of formulating the problem this way is that this model can straightforwardly accommodate tests between multiple SNPs and all isoforms at once, although we do not pursue that use case in this manuscript.

Finally, instead of performing regressions between each *cis*-SNP and each isoform and then aggregating the information across isoforms from the same gene, one may simply perform a multiple linear regression between each SNP and all isoforms, and then use an F-test to test

whether the SNP is associated with any isoform. In some sense this is the inverse of the model described above. Here, the SNP is the response variable in the regression and the isoforms are the predictors. In the general linear model formulation, we have a stack of simple linear regressions which each have the SNP as the predictor variable and one of the isoforms as the response variable.

More details on each of these methods is presented in the Methods section. We compare the relative merits of these approaches for identifying eQTLs in simulation and real data analyses below.

## Simulation study

We first evaluated the relative performance of the methods described above by performing a simulation study using the genotypes of 87 Yoruban donors from the GEUVADIS [4] dataset. We selected a realistic set of non-zero isoforms and genes by applying kallisto [23] to the GEU-VADIS RNA-seq data for the Yoruban samples to identify and filter out isoforms with near-zero expression levels (S1 Text). For each gene we randomly selected $k \in (1, 2, 3)$ causal SNPs within a 100 kilobase window around the transcription start site. We then simulated expression levels for each isoform via the following model:

$$\mathbf{Y} = \boldsymbol{\beta}\mathbf{X} + \mathbf{U} + \epsilon$$

where $\mathbf{Y}$ is an $(m \times n)$ matrix containing the isoform expression levels across $m$ isoforms for the gene and $n$ samples, $\boldsymbol{\beta}$ is an $(m \times k)$ matrix of effect sizes of each of the $k$ causal SNPs on the $m$ isoforms, $\mathbf{X}$ is a $(k \times n)$ matrix containing the SNPs of the $k$ causal SNPs across $n$ samples, $\mathbf{U}$ is an $(m \times n)$ matrix containing non-*cis* effects on isoform expression, and $\epsilon$ is an $(m \times n)$ noise matrix. The matrix $\mathbf{U}$ represents effects on isoform expression levels that are correlated between both isoforms and samples, but are not due to *cis*-effects. For instance, this could correspond to environmental effects or *trans*-regulation.

We assume that each causal SNP $\mathbf{X}_j$ affects the expression levels of the isoforms independently of one another, so we can simulate

$$\mathbf{Y} = \sum_j \boldsymbol{\beta}_j\mathbf{X}_j + \mathbf{U} + \epsilon$$

We assume, however, that a SNP may affect different isoforms in a correlated manner, e.g. if the SNP is a splicing QTL. Thus we assume that the causal effects of the SNP on the $m$ isoforms is drawn from a multivariate normal distribution,

$$\boldsymbol{\beta}_j \sim \mathcal{N}(\mathbf{0}, \sigma_{g_j}^2 \boldsymbol{\Phi}_j)$$

where $\boldsymbol{\Phi}_j$ is an $(m \times m)$ covariance matrix with diagonal entries equal to 1 and off-diagonal entries between -1 and 1 (S1 Text). The $\sigma_{g_j}^2$ parameter controls the percent of variance of isoform expression level explained by the SNP. In our simulations, the parameter for all SNPs combined, $\sigma_g^2$, is set, and the individual SNP parameters are randomly set such that $\sigma_g^2 = \sum_j \sigma_{g_j}^2$.

We draw $\mathbf{U}$ according to a matrix normal distribution,

$$\mathbf{U} \sim \mathcal{MN}(\mathbf{0}, \mathbf{V} * \sigma_h, \mathbf{W} * \sigma_h),$$

where $\mathbf{V}$ is an $(m \times m)$ covariance matrix between the isoforms, $\mathbf{W}$ is an $(n \times n)$ covariance matrix between the samples, and $\sigma_h^2$ is a tunable parameter controlling the proportion of variance in isoform expression explained by $\mathbf{U}$.

Finally, the noise parameter for each isoform $i$, which is not correlated across isoforms or samples, is drawn according to

$$\epsilon_i \sim \mathcal{N}(\mathbf{0}, \sigma_e^2 \mathbf{I}),$$

where $\sigma_e^2 = 1 - \sigma_g^2 - \sigma_h^2$ and $\mathbf{I}$ is the $n$-dimensional identity matrix.

We simulated expression levels for each isoform of every gene in the GEUVADIS data with $m = 4$ isoforms according to this model under multiple parameter settings, with $\sigma_g^2 \in \{5\%, 10\%, 15\%, 20\%\}$ and $\sigma_h^2 \in \{0\%, 5\%, 10\%, 20\%, 40\%\}$ (for results with different $m$, see Fig B in S1 Text). We also varied the percentage of covariances between $cis$-effects on isoforms that were negative—in other words, the percent of SNP-isoform effects that would wholly or partially "cancel out" when summed together. The larger this parameter, the worse we expect summation-based methods to perform. We tested five values for this parameter, $negpct \in \{10\%, 30\%, 50\%, 70\%, 90\%\}$. We set the default parameters to be $\sigma_g^2 = 0.05$, $\sigma_h^2 = 0.0$, and $negpct = 0.5$, and then performed three different experiments in which one parameter was allowed to vary within the settings listed above and the other two were fixed at their default values. Regardless of the $\sigma_g^2$ setting, we set each gene to have $\sigma_g^2 = 0$ with 50% probability, so that both power (to identify eGenes) and false discovery control could be evaluated. We ran each method using these simulated expression levels and the Yoruban GEUVADIS genotypes as input, with the false discovery threshold set to 10% (Fig 2). Below, we refer to p-value aggregation methods applied prior to the permutation test with the suffix "-perm", and when applied to QTLtools run on individual isoforms (i.e. after the permutation test) we refer to them with the suffix "-qtltools-iso" (see Table 1 for reference).

In most settings, most methods had empirical false discovery rates (FDR) below 10%, indicating the effectiveness of the permutation test and FDR approach for controlling false discoveries, and isoform-aware methods outperformed approaches that simply sum or average the isoform expression levels, with Fisher-perm generally being the most powerful calibrated method. The F-test was the most powerful method that did not have inflated FDR in any simulation setting tested. Isoform-aware approaches had roughly 70–80% power when $\sigma_g^2 = 0.1$ and roughly 95% power when $\sigma_g^2 = 0.2$. We note some interesting trends and exceptions below.

First, min-qtltools-iso generally had an inflated FDR of around 20%. This approach can be summarized as counting a gene as an eGene if a SNP is significantly associated with any of the isoforms for that gene. Researchers should be aware of this bias if they intend to apply this approach. The Cauchy combination test performed similarly whether applied before or after the permutation test, while Fisher's method was more powerful when applied before the permutation test.

Second, as expected, approaches that aggregate isoforms by summing or averaging their expression levels were generally less powerful than isoform-aware methods. The underlying cause of this can be observed in the plot where the percent of negatively correlation SNP-isoform effects ("$negpct$") is varied. When $negpct = 0.1$, these approaches have power comparable to some of the isoform-aware approaches, but as $negpct$ is increased their power drops dramatically, reaching nearly zero when $negpct = 0.9$. This stands to reason since negatively-correlated $cis$-effects will partially or fully cancel out when summed together or averaged. The performance when summing versus averaging isoform expression levels was nearly identical, as expected, since these approaches only differ by a constant. Additionally, with $negpct = 0.5$, as the number of isoforms per gene increased, the power of isoform-aware methods increased, while that of sum or mean approaches did not, or even slightly decreased (Fig B in S1 Text).

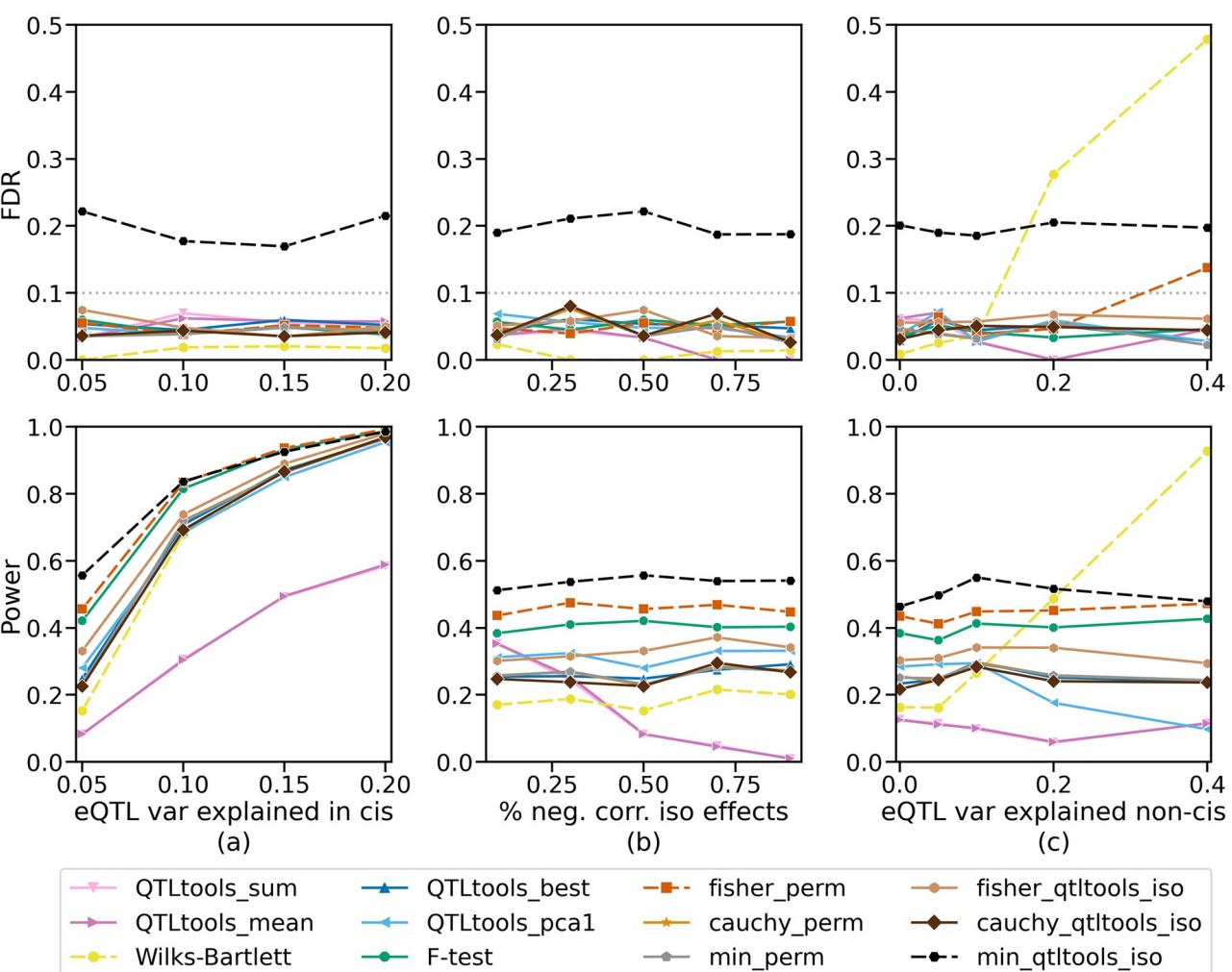

**Fig 2.** Simulation results with varying (a) percent variance of isoform expression explained by causal SNPs in *cis*; (b) percent of *cis*-eQTL effects that are negatively correlated between the different isoforms for a gene; (c) percent variance of isoform expression explained by non-*cis* effects that are correlated across samples and isoforms. Varying parameter shown on the X-axis with empirical False Discovery Rate (FDR; top row) or Power (bottom row) on the Y-axis. When not being varied, these parameters were fixed at 0.05, 0.5, and 0.0, respectively. Dashed lines indicate methods which had inflated empirical FDR in at least one simulation setting. The method types are organized by color: Gene-level methods are in purple/pink, QTLtools grouped permutation methods are in dark/light blue, Wilks-Bartlett is in yellow, F-test is in green, and p-value aggregation methods are in oranges, browns, gray, and black.

This is likely due to the larger number of isoforms giving more opportunities for isoform-aware methods to identify true positive effects, while simultaneously creating more opportunities for these isoforms to "cancel out" when negatively correlated, harming sum or averaging approaches. We note, however, that when genes have many isoforms in real data, many of these isoforms may be lowly expressed, so this trend in our simulations may not translate precisely to real data.

Third, when the non-cis heritability, $\sigma_h^2$ was increased, several trends emerged. Most strikingly, the Wilks-Bartlett approach exhibited a highly inflated FDR, eventually predicting almost every gene to be an eGene. This result follows since a non-diagonal **U** violates the Wilks-Bartlett assumption that regression residuals will be uncorrelated across samples, as **U** induces this correlation. Although less dramatic, Fisher-perm also appears to have a slightly

inflated FDR when $\sigma_h^2 = 0.4$, which may be because of the method's assumption that the p-values from individual isoform regressions are uncorrelated, which is known to occasionally lead to inflated FDR [29]. Finally, QTLtools-pca1 loses power as $\sigma_h^2$ is increased. When $\sigma_h^2 = 0$, QTLtools-pca1 and QTLtools-best have similar power, but as $\sigma_h^2$ is increased to 0.4, QTLtools-pca1 loses power until it is comparable to the sum and averaging approaches. This is likely because the first principal component becomes distorted by the complex covariance structure between isoforms induced by **U**.

Fourth, the methods that did not exhibit inflated FDR or reduced power when any of the parameters were varied generally followed a consistent ranking in terms of power across most settings tested. The F-test approach was generally the most powerful of these approaches, followed by Fisher-qtltools-iso. Next, QTLtools-best, min-perm, Cauchy-perm, and Cauchy-qtltools-iso generally performed quite similarly. When not inflated, Fisher-perm was generally the most powerful method, albeit closely followed by the F-test. The Wilks-Bartlett approach had similar power to the less powerful isoform-aware approaches when its FDR was not inflated. QTLtools-pca1 had similar power to QTLtools-best when its power was not deflated.

Overall, the results suggest that Fisher-perm is a powerful approach that usually remains calibrated but can exhibit inflated FDR under strong non-*cis* effects, while the F-test and Fisher-qtltools-iso appeared to be the most powerful methods that were robust to all parameter settings in terms of FDR. When using QTLtools, the "grp-best" permutation approach (as it is called in the software) appears to be a robust and somewhat powerful method that is preferable to the "grp-pca1" and "grp-mean" settings.

## Application to GEUVADIS data

We next evaluated the performance of these methods on real expression data. Using the same 87 Yoruban samples from the GEUVADIS dataset [4] described in the simulation study, we quantified the normalized transcript-per-million (TPM) expression level of each isoform using kallisto [23] and DESeq size factors [30] (S1 Text). Isoforms with 5 or fewer reads mapped for at least 50% of the samples were discarded. The methods were run on the remaining isoform and genotype data. The corresponding quantile-quantile (Q-Q or QQ) plot is shown in Fig A in S1 Text.

The findings on this data largely recapitulated those of the simulations, and strikingly resembled those of the simulation where $\sigma_h^2 = 0.4$ and *negpct* = 50%. The Wilks-Bartlett approach identified 9364 eGenes, much more than the next-greatest approach (4500), seemingly suggesting a large number of false discoveries. The min-qtltools-iso approach identified the next most, with 4500 eGenes, likely also due to inflated FDR. Next was Fisher-perm, with 4023 eGenes. The remaining isoform-aware approaches identified 2342 to 3003 eGenes, with Fisher-qtltools-iso identifying the most, followed by the F-test (2661). QTLtools-sum, QTLtools-mean, and QTLtools-pca1 identified 1552, 1294, and 1432 eGenes, respectively. We note that the original GEUVADIS study identified 501 eGenes in the Yoruban samples [4]. We attribute this difference to several factors in the analysis pipeline, including our use of kallisto [23] and DESeq size factors [30] to quantify expression levels versus the original study's use of summed transcript reads per kilobase million (RPKM, which is now recommended against in current practice [10, 31]) and the original study's use of Matrix eQTL [8] to call eGenes.

A common alternative to summing isoform-level quantifications as we did for QTLtools-sum is to use read counting or alignment methods such as featureCounts [25], HTSeq [32], or STAR [33] to quantify gene expression levels. We also ran QTLtools on gene read counts obtained from featureCounts. This approach identified more eGenes compared to the QTLtools-sum approach (1718 versus 1552) but substantially fewer than the isoform-aware methods, which each found 2342 or more eGenes.

We next analyzed the overlap of eGenes found by each method, excluding methods that had inflated empirical FDR in any simulation setting as well as non-isoform-aware methods. These methods (QTLtools-best, F-test, fisher-qtltools-iso, cauchy-qtltools-iso, cauchy-perm, min-perm) found 3,711 total genes, of which 1,876 were shared by all methods, indicating substantial concordance. Fisher-qtltools-iso had the most unique eGenes with 368, followed by F-test with 316, QTLtools-grpbest with 65, cauchy-qtltools-iso with 31, and cauchy-perm and min-perm with 3 each.

Given that the Wilks-Bartlett approach seemingly had a highly-inflated eGene count and QTLtools-pca1 had a count similar to the sum and average approaches, which strikingly resemble the results in our simulations with high $\sigma_h^2$, this suggests that non-*cis* effects that are correlated across isoforms and individuals, possibly including trans-regulation and environmental effects, are prevalent in real datasets. Another resemblance between the real data and simulation findings was that Fisher-qtltools-iso and the F-test identified the most eGenes among the non-inflated isoform-aware approaches. Interestingly, it appears that the F-test was less powerful relative to the other isoform-aware approaches in GEUVADIS than it was in simulations.

Additionally, the results corroborated our simulation findings that sum- and mean-based approaches will identify fewer eGenes than isoform-aware methods. Fig 1 shows an example gene that isoform-aware methods identified but QTLtools-mean did not. Clearly, in this example, the two isoforms for the gene are regulated in opposite directions by the SNP, such that summing the expression levels of the isoforms loses some of the signal. In general, we found that isoform-aware methods tended to find eGenes with isoforms that were genetically regulated in opposite directions, as expected. For example, we examined the eGenes with multiple isoforms identified by both the F-test and QTLtools-sum, versus those identified only by the F-test. We calculated the number of eGenes whose isoforms have opposite-sign associations with the eQTL, meaning that the eQTL had a positive-sign effect estimate on at least one isoform and a negative-sign estimate on at least one isoform. We found that 88.3% (1221 of 1383) of such eGenes found by the F-test but not QTLtools-sum have isoforms with opposite-sign associations with the eQTL. Meanwhile, of the multi-isoform eGenes found by both methods, only 50.1% (326 of 651) have opposite-sign associations with the eQTL. If we require the opposite-sign isoform associations to be marginally significant at $p < 0.05$, then 47.9% (663 of 1383) of the F-test-exclusive eGenes have significant opposite-sign associations, while only 22.1% (144 of 651) of eGenes found by both methods fit this criterion.

Given that isoform-aware methods are likely to identify some splicing QTLs (sQTLs) in addition to traditional eQTLs, we hypothesized that the frequency distribution of QTL locations relative to the gene body would differ between isoform-aware methods and QTLtools. We found that two different isoform-aware methods, Fisher's method and QTLtools-best grouped permutations, had relatively similar QTL frequency distributions while QTLtools-mean was rather different (Fig 3). eQTLs identified by QTLtools-mean were skewed towards the immediate vicinity of the transcription start site (TSS), while the isoform-aware methods identified relatively more eQTLs towards the end of the gene body and beyond the gene body. Interestingly, Li et al [34] previously reported that eQTLs are enriched near the TSS of the gene they regulate, while sQTLs are enriched throughout the gene body, particularly within introns. We also noticed from the Li et al findings that, outside of the gene body, most eQTLs were found to be immediately before the TSS while more sQTLs were found after the gene body relative to eQTLs. Our findings largely corroborate these trends, suggesting that isoform-aware methods are picking up in sQTLs in addition to eQTLs. Surprisingly, isoform-aware methods only had a slightly higher percentage of eQTLs within introns (31.6% for Fisher's

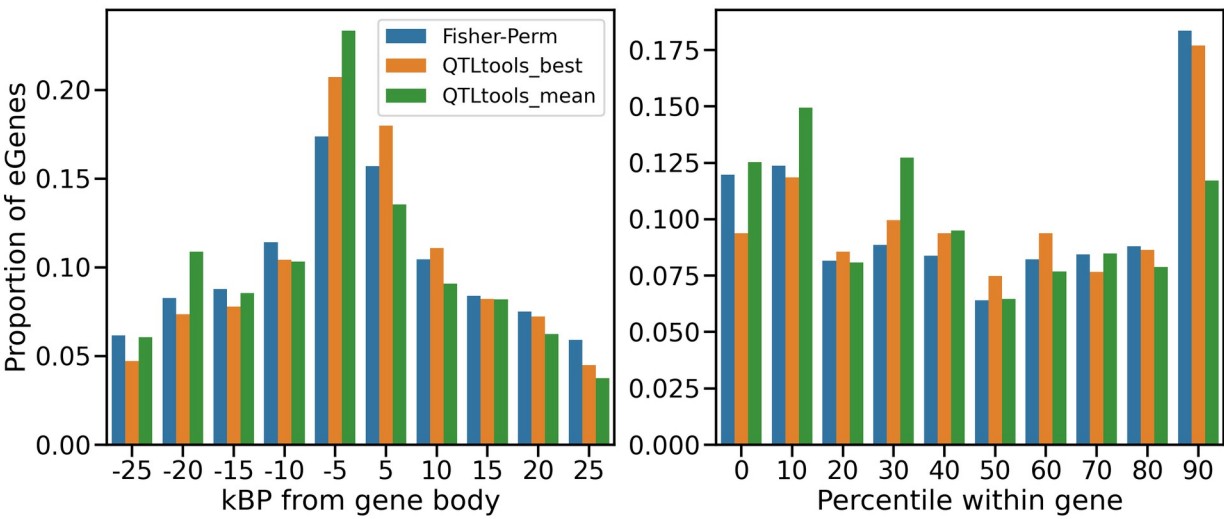

**Fig 3.** Counts of eGenes found by each method, binned by distance from transcription start site (TSS) for eQTLs found outside the gene body (left); or by percentile position within the gene body for eQTLs found within the gene body (right), where 0 is the start of the gene and 100 is the end of the gene.

method and 32.2% for QTLtools-best) than did QTLtools-mean (28.7%). This may be because the isoform-aware methods are finding both eQTLs and sQTLs, not just sQTLs.

The running times of each of the methods are reported in the S1 Text. In general, the methods are difficult to compare directly due to their different implementations and pipelines. We found that the majority of the runtime is taken up by running SNP-isoform or SNP-gene regressions for each of the permutation replicates, rather than the specific testing method employed. Because QTLtools implements these basic regressions and permutations more effectively than the methods we implemented ourselves, their runtimes were generally much faster.

## Application to GTEx Data

As another evaluation on more recent real data, we applied the methods to the GTEx v8 data [6]. We decided to use thyroid samples because this tissue had among the most eGenes of any tissue in the GTEx v8 data [6]. We preprocessed the data largely the same way as described in the Supplementary Text of the GTEx v8 paper [6]—the expression data were normalized with TMM [35], filtered for low expression, and then inverse-normal transformed. The main differences were that i) we used only European samples to avoid potential population stratification; ii) we performed low-expression filtering on isoform expression levels rather than aggregated gene expression levels; iii) normalization was applied to isoform expression levels rather than aggregated gene expression levels (S1 Text). We then performed association testing with the methods described above, controlling for covariates such as principal components and batches as described in the GTEx v8 Supplementary Text [6].

Our results largely recapitulated the trends observed in our other experiments. The fewest eGenes were identified by QTLtools-pca1 (9659) and QTLtools-mean (10241). Wilks-Bartlett identified 16975 eGenes, considerably more than any other method, and was once again likely inflated. The next most was Fisher-perm with 13825, followed by min-qtltools-iso with 13728, then F-test with 13271. Fisher-qtltools-iso, Cauchy-qtltools-iso, QTLtools-best, Cauchy-perm, and min-perm identified 12644, 12568, 12519, 12410, and 12395 eGenes, respectively. QTLtools-mean is analogous to FastQTL, which the method used in the GTEx eQTL analysis,

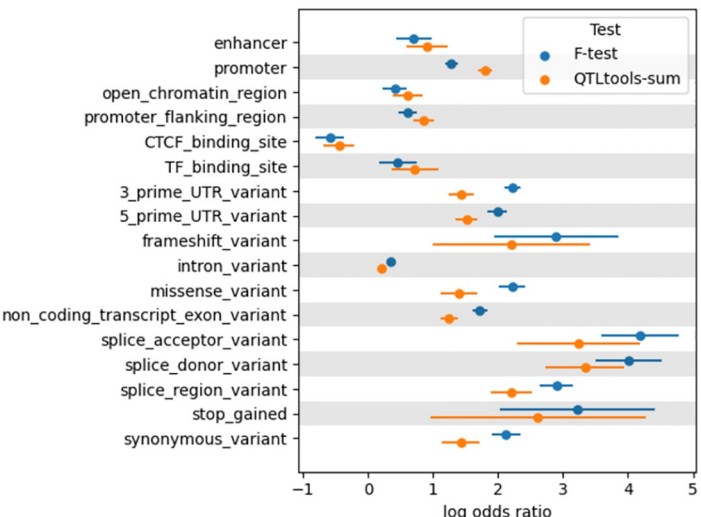

**Fig 4. Functional enrichment for various genomic annotations for eQTLs identified in the GTEx thyroid data by an isoform-aware method (the F-test) and a gene-level method (QTLtools-sum).**

in that QTLtools-mean is essentially FastQTL run on gene-level expression data that is obtained by averaging isoform expression levels rather than summing them. However, QTLtools-mean found fewer eGenes than the GTEx v8 eQTL analysis on the thyroid (13477), which was expected given the differences in the analysis pipeline described above. Overall, these results recapitulate our findings that accounting for isoform-level expression increases power to identify eGenes and our findings about the relative power of each isoform-aware method.

We once again analyzed the overlap of eGenes found by each method, excluding methods that had inflated empirical FDR in any simulation setting as well as non-isoform-aware methods. These methods (QTLtools-best, F-test, fisher-qtltools-iso, cauchy-qtltools-iso, cauchy-perm, min-perm) found 13,917 total eGenes, of which 11,808 were shared by all methods, indicating high concordance. F-test had the most unique eGenes with 738, followed by fisher-qtltools-iso with 120, QTLtools-grpbest with 87, cauchy-qtltools-iso with 32, cauchy-perm with 3, and min-perm with 0 (no unique eGenes).

We then performed functional enrichment analysis using torus [36] applied to eQTLs identified by an isoform-aware method (F-test) and a gene-level method (QTLtools-sum) given genomic annotations from the GTEx v8 dataset [6] (Fig 4). F-test and QTLtools-sum had relatively similar functional enrichment profiles, and were both fairly similar to the eQTL enrichment profile found in the GTEx v8 paper [6]. However, the isoform-aware F-test was more enriched for splicing-related categories, suggesting that some QTLs identified by isoform-aware methods that are not identified by gene-level methods may be related to splicing.

## Discussion

In this paper, we examined several approaches for utilizing isoform expression information in eQTL mapping. We demonstrated that the common approach of summing or averaging the isoform expression estimates together results in a loss of power to identify eGenes relative to approaches that utilize isoform-specific information in situations where a SNP regulates isoforms in possibly different directions. Though we did not explicitly evaluate it here, a related scenario in which this approach may lose power is when the expression level of one isoform is

much higher than that of the other isoforms for the same gene, such that the highly-expressed isoform masks genetic regulation of the low-expressed isoforms when the expression levels are added together [11].

We examined four classes of approaches to utilizing isoform-specific information: QTLtools' grouped permutation schemes, p-value aggregation, general linear models, and multiple regression. We also examined whether it is preferable to apply p-value aggregation methods to permutation-adjusted p-values generated by running QTLtools [17] on individual isoforms, or to apply these methods to raw isoform-level regression p-values and subsequently compute the permutation-adjusted p-value. While no single approach dramatically outperformed the others, Fisher's method [18] applied prior to the permutation test ("Fisher-perm") generally had the highest power among methods that did not have dramatically inflated False Discovery Rate (FDR) in simulations, followed by the multiple regression F-test and Fisher's method applied to QTLtools isoform-level results ("Fisher-qtltools-iso"), with most of the remaining methods performing very similarly to one another. However, the Wilks-Bartlett method and, to a lesser extent, Fisher-perm, exhibited inflated FDR when substantial non-*cis* effects were simulated. Among the methods that did not exhibit inflated FDR in any simulation setting, the F-test generally had the highest power in simulations and found the most eGenes in the GTEx data, while Fisher-qtltools-iso found the most eGenes in the GEUVADIS data.

We observed conflicting and inconclusive answers as to whether it is preferable to apply p-value aggregation approaches before or after the permutation test. Fisher-perm exhibited slight FDR inflation in some settings, as discussed above, but Fisher-qtltools-iso did not. On the other hand, min-perm did not exhibit inflated FDR but min-qtltools-iso did, cautioning against the approach of calling an eGene if a SNP is significantly associated with any of the gene's isoforms. The Cauchy combination test [19] did not exhibit inflated FDR and had similar power in both settings. Numerous p-value aggregation methods exist, including those that weight different p-values or attempt to account for dependencies between p-values [29, 37–40]. For the sake of brevity, we were not able to explore all of these options, and it remains to be seen whether they would substantially improve performance.

Results on the GEUVADIS data strongly resembled our simulation setting with strong non-*cis* effects and roughly half of SNP-isoform effects being negatively correlated. In particular, Wilks-Bartlett appeared to yield a highly inflated eGene count, while QTLtools-pca1 identified a similar number of eGenes to the sum and average approaches. Results on the GTEx data had similar trends. Real gene expression data is much more complicated than our simulations can capture, with a variety of molecular processes and confounding signals involved, and certainly it does not match any particular parameter setting in our simulation. However, these findings do suggest that many eGenes are substantially impacted by non-*cis* effects, in line with previous research that suggests that *trans*-effects on gene expression account for more gene expression heritability than *cis*-effects across the genome [41–46]. Methods for identifying eQTLs and simulations assessing these methods should take this factor into account.

Examining the GEUVADIS results further, we found that the eQTLs identified by QTLtools-mean were skewed towards the transcription start site (TSS) of the genes they regulated, while isoform-aware methods found relatively more eQTLs towards the end of the gene body and beyond the gene body. As these trends largely matched those described by Li et al [34] comparing eQTL locations to splicing QTL (sQTL) locations, these findings suggest that some of the QTLs identified by isoform-aware methods are likely sQTLs. Additionally, in a functional enrichment analysis of the eQTLs identified in the GTEx data, eQTLs identified by the isoform-aware F-test were more enriched for splicing-related categories than eQTLs identified by the gene-level QTLtools-sum approach. In this sense, isoform-aware methods likely identify a broader set of biological signals than traditional eQTL mapping identifies.

This point about a potentially broader set of signals warrants further discussion. Traditional eQTL mapping methods typically seek to identify SNPs associated with the "gene-level" expression of genes—i.e. the sum of the isoform expression levels corresponding to a gene. While this goal may be interesting, we think it is valuable to develop methods with the broader goal of identifying genetic regulation of expression. Accordingly, we define an eGene as a gene with at least one isoform whose expression level is regulated by a *cis*-SNP. This may include effects specific to one or a subset of isoforms, such as alternative splicing or polyadenlyation. An alternative approach is to consider specific kinds of regulatory QTLs—e.g. "gene-level" eQTLs and sQTLs—separately. Several methods have been developed that are dedicated to finding sQTLs [47–49]. If one wanted to explicitly separate traditional eQTLs from sQTLs, one could consider running the QTLtools-sum or QTLtools-mean approach for eQTLs and one of the sQTL methods from sQTLs. This has the benefit of delineating specific kinds of regulatory QTLs, but has the drawbacks of requiring multiple analyses, exacerbating multiple testing burden, and not capturing other types of regulatory QTLs. These two approaches may be considered complementary, as each have their benefits and drawbacks.

For simplicity, in this manuscript, we focused on a specific problem: detecting cis-eQTLs and eGenes given quantified isoform expression levels generated by a method such as kallisto [23] or Salmon [24]. A similar approach that we do not consider here is using exon quantifications rather than isoform quantifications, which can also be computed using kallisto or Salmon. Alternatively, differential exon usage methods like DEXSeq [50] could be adapted for eQTL/eGene analysis. While many reads will be compatible with multiple isoforms, presenting a challenge for estimating individual isoform quantities, this is generally less of an issue for exon quantification. While this is possible, it has been noted that these methods suffer from other issues such as not utilizing full information of junction reads [51]. Another option is to work with read count data obtained from a method such as featureCounts [25] or HTSeq [32], which could be either on the gene or exon level. We found in our analysis of GEUVADIS data that QTLtools run on featureCounts read counts obtained slightly more eGenes than QTLtools run on summed kallisto isoform quantifications, albeit still substantially fewer eGenes than isoform-aware methods. Reliably quantifying isoform expression can be challenging, particularly for lowly expressed isoforms [52]. In such cases, it may be preferable to work with simpler count-based data such as that obtained from featureCounts. Since these counts will be relatively low, it may be preferable to apply a Poisson or negative binomial model and test for association using a generalized linear model rather than ordinary least squares. This is an interesting direction for future research.

We also note that there are other methods for finding molecular QTLs, including eQTLs, that involve different computational pipelines. For example, WASP [53] and RASQUAL [54] are QTL-finding methods that include mapping stages in their pipeline that are designed to identify allele-specific reads and reduce the effects of several read mapping biases. These methods increase QTL mapping power relative to simple linear regression at the cost of substantially slower computational speed [54], and may be preferable in some contexts. These methods do not currently account for isoform-level effects. Because these methods involve substantially different pipelines and are not trivial to extend to multiple-isoform contexts, we do not examine them here.

One other approach that we considered but did not evaluate due to its substantial differences with the evaluated methods is aggregating information from multiple SNPs in cis to increase power to identify eGenes, rather than testing each SNP in cis individually. Some methods for doing this are implemented in QTLtools [17], but we are not aware of any extensive evaluations of the performance of these methods relative to other possible approaches.

Another dimension in which this problem can be approached differently is in the choice of FDR control. In this manuscript we used the beta-distributed approximate permutation test (Methods) along with Storey's Q-value [26], as suggested by FastQTL [9], to control FDR. Alternative approaches include the overall FDR [15, 16, 55] and hierarchical FDR [56, 57]. We found the FastQTL approach to be effective for controlling FDR, and for simplicity, we do not evaluate these alternatives here. However, one disadvantage of this approach is that it focuses on gene-level testing, and does not obviously suggest a principled way to identify the isoforms that drove the gene-level signal (apart from identifying individual isoform-level p-values that are significant). Methods such as stageR [15] offer an alternative approach to FDR control that does allow for identifying the driver isoforms. In the stageR method, approaches that we evaluated in this manuscript such as the F-test or p-value aggregation form an initial "screening stage," which is then followed by a "confirmation stage" in which the individual (isoform-level) tests are re-assessed and controlled for FDR or Family-Wise Error Rate. It would be interesting to compare the stageR and FastQTL (and other) approaches in future work.

The findings in this paper may be relevant to other contexts. For example, it would be interesting to compare some of these approaches for trans-eQTL mapping. Similar approaches could be developed for different types of molecular QTLs for which it is appropriate to account for hierarchical information. Another possibility is extending the approaches presented in this paper to other contexts in which hierarchical information is available. Most straightforwardly, these approaches could be applied directly to exon expression levels rather than isoform expression levels. Other applications, such as identifying QTLs associated with the expression level of any gene in a gene set, or with genes whose expression levels are correlated due to their involvement in similar pathways within a co-expression network, may also be interesting directions for future work. Finally, similar approaches may be used to more-powerfully identify expression or other molecular traits that mediate genetic effects on complex traits and diseases. Indeed, recent work has shown that isoform-level modeling increases power in Transcriptome-Wide Association Studies (TWAS) [58]. Utilizing hierarchical information could potentially lead to substantial power increases across a variety of biological association mapping problems, providing further opportunities for developing biological insights.

## Methods

### P-value aggregation methods

Methods for aggregating p-values from multiple hypothesis tests have been studied extensively, and there are too many methods to cover comprehensively here. We focus on one naive method—simply using the minimum p-value—and two more well-founded and popular approaches, the classic Fisher's method [18] and the currently-popular Cauchy aggregation test, also called "ACAT" [19, 27].

Fisher's method aggregates the p-values into a chi-squared statistic:

$$\chi^2_{2k} = -2\sum_{i=1}^{k}\log(p_i),$$

where $k$ is the number of studies, $p_i$ is the p-value from study $i$, and $\chi^2_{2k}$ is the chi-squared distribution with $2k$ degrees of freedom. The basic principle is that a p-value is uniform under the null, the negative log of a $Unif(0, 1)$ random variable is exponentially distributed, and multiplying an exponential random variable by 2 yields a chi-squared random variable with two degrees of freedom. Summing $k$ such variables yields a chi-squared random variable with $2k$ degrees of freedom.

Fisher's method is asymptotically optimal when p-values are independent [59, 60] but fails to take into account correlations between the p-values. If they are correlated, Fisher's method may result in anti-conservative aggregated p-values. Numerous extensions to Fisher's method have been proposed to deal with dependent p-values or weighting of p-values [29, 37–40].

One recent popular alternative is the Cauchy combination test [19, 27], which is based on the observation that Cauchy statistics can be used to correct for possible dependence structures between p-values. The Cauchy combination test is defined as

$$T = \sum_{i=1}^{k} \omega_i tan([0.5 - p_i]\pi),$$

where $\omega_i$ are weights on the p-values. If the observed test statistic is $T = t_0$, the p-value can be approximated as

$$p = \frac{1}{2} - \frac{arctan(t_0)}{\pi}.$$

We use the Cauchy combination test as a representative modern p-value aggregation approach.

## Wilks-Bartlett test

Let $n$ be the number of samples and let $m$ be the number of isoforms for the gene being tested. Let $\mathbf{Y}$ be the $(n \times m)$ matrix of isoform expression levels. Let $\mathbf{X}$ be the $(n \times 1)$ SNP dosage vector for any SNP in *cis*. Then we model

$$\mathbf{Y} = \mathbf{X}\boldsymbol{\beta} + \epsilon,$$

where $\boldsymbol{\beta}$ are effect sizes of the SNP on each isoform and $\epsilon$ is the $(n \times m)$ error matrix, which is assumed independent across samples. The hypothesis we want to test is:

$$H_0 : \boldsymbol{\beta} = 0$$
$$H_1 : \boldsymbol{\beta} \neq 0.$$

That is, under the null, a SNP has zero effect on each isoform.

A likelihood ratio test can be constructed [20] for this which results in the Wilks' lambda-distributed [21] test statistic

$$\Lambda^{2/n} = \frac{|\hat{\Sigma}|}{|\hat{\Sigma}_i|},$$

where $\hat{\Sigma} = \frac{1}{n}\hat{\epsilon}^T\hat{\epsilon}$, with $\hat{\epsilon}$ being estimated via the ordinary least squares (OLS) estimate for $\boldsymbol{\beta}$, and $\hat{\Sigma}_i$ is the equivalent matrix for the intercept-only model.

Bartlett [20, 22] showed that the modified statistic

$$-\left[n - r - 1 - \frac{1}{2}(m - r + q + 1)\right]\log\left(\frac{|\hat{\Sigma}|}{|\hat{\Sigma}_i|}\right)$$

has approximately a $\chi^2_{m(r-q)}$ distribution for large $n$, where $r$ is the total number of regressors and $q$ is the number of regressors not being tested for zero effect. Thus, in our case, $r = 1$ (since

we test one SNP at a time) and $q = 0$, so the statistic simplifies to

$$-\left[n - 2 - \frac{1}{2}m\right]\log\left(\frac{|\hat{\Sigma}|}{|\hat{\Sigma}_i|}\right)$$

and has an approximate $\chi_m^2$ distribution.

## Multiple regression and F-test

Let $n$ be the number of samples and let $m$ be the number of isoforms for the gene being tested. Let $\mathbf{Y} = (\mathbf{Y}_1, \ldots, \mathbf{Y}_m)$ be the $(n \times m)$ matrix of isoform expression levels. Let $\mathbf{X}$ be the $(n \times 1)$ SNP dosage vector for any SNP in *cis*. We then perform the multiple regression

$$\mathbf{X} = \mathbf{Y}\boldsymbol{\beta} + \epsilon_f.$$

We also perform the intercept-only regression. Let $RSS_f$ represent the residual sum of squares computed from the full model above, and let $RSS_i$ represent the residuals sum of squares computed from the intercept-only model. We can then perform an F-test for whether the full model predicts $\mathbf{X}$ better than the intercept-only model. The F-statistic will be

$$F = \frac{(RSS_i - RSS_f)/(d_i - d_f)}{RSS_f/(n - d_f)},$$

where $d_i = n - 1$ is the degrees of freedom in the intercept-only model and $d_f = n - m - 1$ is the degrees of freedom in the full model. An F-test is then performed using this statistic to compute the gene-level p-value, with the two degrees of freedom parameters being $d_f$ and $m - 1$.

## Beta-approximated permutation test and False Discovery Rate control

When performing eQTL mapping, multiple SNPs are associated with each phenotype (a gene or isoform expression level) and multiple phenotypes are tested across the genome. Both of these multiple testing burdens must be corrected for. FastQTL [9] introduced a scheme for doing so, which was later incorporated into the QTLtools [17] software and which we emulate for each method discussed in this paper.

SNPs in the same region are correlated via linkage disequilibrium, and a gold standard approach for adjusting for multiple testing with correlated tests is permutation testing. Permutation testing involves generating $R$ rearrangements of the data points, computing the hypothesis test for each of the R permutations as well as the original data, and then comparing the p-value for the original data with those obtained using the permuted data. The permutation test p-value will be

$$\frac{r + 1}{R + 1}$$

where $r$ is the number of permuted-data p-values that are less than the p-value obtained from the original data. An issue arises from this: it is computationally infeasible to compute extreme p-values using this approach. For example, obtaining a p-value of $10^{-6}$ requires computing one million permutated versions of the data and running the hypothesis test on each of them.

This computational burden can be avoided by using a Beta distribution to approximate the tails of the distribution of permuted p-values, thus allowing the quick computation of permutation test p-values [9]. This is because p-values are uniformly distributed under the null hypothesis, and the order statistics of independent uniformly-distributed random variables are Beta-distributed [61]. Thus, the p-values from a permutation test are also Beta distributed [62].

In our case, because the SNPs are not independent, the p-values from standard regressions are not independent. Thus, the parameters of the Beta distribution are fit via a maximum likelihood approach [63] using a limited (about 100–1000) number of permutations to generate a null set of p-values [9].

The authors of QTLtools [17] extended this methodology to accomodate "grouped" phenotypes, such as multiple isoforms from the same gene. We refer to their paper for more details, but briefly, the idea is to permute all isoforms from a given gene at once to generate a null distribution and fit the Beta distribution parameters. Then each SNP-isoform pair's permutation p-value is computed, and the SNP-isoform pair with the strongest p-value is reported for each gene. As an alternative way to aggregate multiple phenotypic signals, they suggested computing loadings on the first principal component of the isoforms, and then performing the regular permutation scheme described above on this. Finally, the isoform expression levels can simply be averaged, which is proportional to simply summing the isoforms.

Once the permuted p-values are obtained for each gene, standard False Discovery Rate (FDR) methods can be applied across genes. The authors of FastQTL recommend Storey and Tibshirani's q-value [26], which is the method we also use in this paper.

There are alternative approaches to this two-level multiple testing problem, including the overall FDR [15, 16, 55] and hierarchical FDR [56, 57]. We found the approach outlined above to be effective for controlling FDR, and for simplicity, we do not evaluate these alternatives here.

## Supporting information

**S1 Text. Supplementary Materials for: Accounting for Isoform Expression Increases Power to Identify Genetic Regulation of Gene Expression.** Details on data preprocessing, software settings, and runtime analysis; quantile-quantile plot of GEUVADIS results; simulation results using genes with different numbers of isoforms.
(PDF)

## Acknowledgments

The authors would like to thank Drs. Arjun Bhattacharya, Bogdan Pasaniuc, and Michael Gandal for their helpful discussion and feedback.

## Author Contributions

**Conceptualization:** Nathan LaPierre, Harold Pimentel.

**Data curation:** Nathan LaPierre, Harold Pimentel.

**Formal analysis:** Nathan LaPierre.

**Funding acquisition:** Harold Pimentel.

**Investigation:** Nathan LaPierre, Harold Pimentel.

**Methodology:** Nathan LaPierre, Harold Pimentel.

**Project administration:** Nathan LaPierre, Harold Pimentel.

**Resources:** Nathan LaPierre, Harold Pimentel.

**Software:** Nathan LaPierre.

**Supervision:** Harold Pimentel.

**Validation:** Nathan LaPierre.

**Visualization:** Nathan LaPierre.

**Writing – original draft:** Nathan LaPierre, Harold Pimentel.

**Writing – review & editing:** Nathan LaPierre, Harold Pimentel.

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
