## [Decision Letter · Decision Letter 0]

27 Nov 2023

Dear LaPierre,

Thank you very much for submitting your manuscript "Accounting for Isoform Expression Increases Power to Identify Genetic Regulation of Gene Expression" for consideration at PLOS Computational Biology. As with all papers reviewed by the journal, your manuscript was reviewed by members of the editorial board and by several independent reviewers. The reviewers appreciated the attention to an important topic. Based on the reviews, we are likely to accept this manuscript for publication, providing that you modify the manuscript according to the review recommendations.

Two reviewers have provided comments. Both reviewers find the manuscript insightful and a substantial research contribution. I agree. Both reviewers make a number of thoughtful points, which should be addressed to improve the manuscript.

Sincerely,

Jeffrey Peter Townsend, Ph.D.

Guest Editor

PLOS Computational Biology

Ilya Ioshikhes

Section Editor

PLOS Computational Biology

Two reviewers have provided comments. Both reviewers find the manuscript insightful and a substantial research contribution. I agree. Both reviewers make a number of thoughtful points, which should be addressed to improve the manuscript.

Reviewer's Responses to Questions

**Comments to the Authors:**

Reviewer #1: The authors detail four approaches to increasing detection power for regulatory variants affecting transcript abundance, using isoform level information. The paper is cleanly written and provides a useful reference for quantitative researchers.

Major comments:

1. For the eGenes detected by isoform-aware methods and not by others, did these tend to have up/down patterns among the isoforms, such that gene-level signal would be masked through cancellation?

2. The authors treat isoform expression as observed variables in a matrix Y. Do the author have consideration or recommendation for working with count data? Additionally, is there any necessary consideration of the fact that counts are not observed from short read RNA-seq, but inferred from an upstream algorithm? Does performance in the simulation depend on the number of isoforms?

3. The simulation mostly focuses on gene-level detection via rejection of null hypotheses. What about post-hoc identification of related isoforms, as in reference [15]? Which of methods are most easy to incorporate into a pipeline that also identifies affected isoforms, while controlling for error in the post-hoc step (again, as in reference [15]).

4. Often genetic variants modifying expression level of genes are also of interest as mediators for downstream traits, as the authors note in the introduction, "Many loci identified in GWAS lie in non-coding regions of the genome, so their functional role is not immediately clear, though it is expected that many such loci play a role in gene regulation Molecular Quantitative Trait Loci (QTL) mapping studies help address this challenge." How may isoform expression be implicated as mediators of downstream GWAS traits? What considerations from this work may extend to the subsequent question of involvement in GWAS traits.

Minor comments:

1. Figure 2, it would be useful to distinguish one way or another the methods presented here according to the four major groupings presented in the main text: traditional, grouped permutation tests, general linear model, multiple regression. In the caption the authors have Wilks-Bartlett, QTLtools sum, QTLtools best, QTLtools pca1, QTLtools mean, F-test, fisher perm, min perm, cauchy perm, fisher qtltools iso, min qtltools iso, cauchy qtltools iso. While these can be associated with the groupings by cross-referencing, it would be beneficial to make this explicit in naming of color groups.

Reviewer #2: This manuscript by LaPierre and Pimentel examines several statistical and computational approaches to identify genes that have at least one isoform whose expression level is regulated by genetic variants. While the problem of identified eGenes is not new this manuscript explains very well the issue that most genes have multiple isoforms and that correctly identifying eGenes is still an open problem. This is an important issue as eGenes may be important for gaining a better understanding of the molecular mechanisms underlying the connection between genetic variants and human traits. The simulations and the two real datasets provide an excellent setting to compare the approaches and understand their advantages and limitations. While some of the approaches are not new, their application for this problem and comparison have not been studied before. Bringing attention to account for different gene isoforms is timely and needed.

I have some relatively minor comments and suggestions to improve the manuscript.

#1) While it is understandable that one needs Kallisto/Salmon isoform level quantification for answering the specific question the authors frame for this paper. In many cases the quantification at the gene level could be done in a simpler way by just aligning the reads with a tool like Hisat2/STAR and then just use a simplified gene model to obtain the read count for the gene. An approach like this should be discussed for context as this has been traditionally done. I understand this may be similar to the QTLtool-sum/mean but it is not exactly the same.

#2) Related to #1,How does the eGENE summary data compare to the summary data from the results consortia provided themselves?

#3) There should also be some discussion on approaches that may map reads to exons first and then QTL map the individual exons rather than the isoforms. This likely will not solve the correlation issues as exons are linked together, but it may deserve some discussion.

#4) It is a bit surprising that permutation after Fisher aggregation (Fisher-perm) is sensitive to the higher level of \\sigma^2_h. I would have expected that the permutation would correct the p-values accordingly. Could there be an issue on how the permutation is done?

#5) It would be useful to see the QQplots for the real datasets results, and perhaps to calculate genomic inflation factors, or compare observed p-value values vs what would be expected at p=0.1 across the multiple methods. The shape of the histogram tail can also influence Storey's q-value method and may also be interesting to look at that.

#6) It would be useful to also provide an estimate of the computational complexity of the approaches. Which approaches would take more time if they require more permutations or more complex computations?

#7) Figure 2, it would be useful to zoom in the FDR y-axis, and do not have that much white space.

**Have the authors made all data and (if applicable) computational code underlying the findings in their manuscript fully available?**

Reviewer #1: Yes

Reviewer #2: Yes

PLOS authors have the option to publish the peer review history of their article (what does this mean?). If published, this will include your full peer review and any attached files.

Reviewer #1: **Yes: **Michael Love

Reviewer #2: No

Figure Files:

Data Requirements:

Reproducibility:

References:

---

## [Editor Report · Decision Letter 1]

23 Jan 2024

Dear LaPierre,

We are pleased to inform you that your manuscript 'Accounting for Isoform Expression Increases Power to Identify Genetic Regulation of Gene Expression' has been provisionally accepted for publication in PLOS Computational Biology.

Best regards,

Jeffrey Peter Townsend, Ph.D.

Guest Editor

PLOS Computational Biology

Ilya Ioshikhes

Section Editor

PLOS Computational Biology

Thank you for your careful and thorough revision in response to reviewer comments. All comments have been satisfactorily addressed, and the resulting manuscript is an excellent contribution to PLOS Computational Biology.

I have one editorial request regarding the writing. In at least two places in the revision text,

This is likely due to …

…. this is generally less of an issue…

as well as e.g. line 48 from the text of the original manuscript,

“...this may not always be the case…"

the demonstrative pronoun "this" is used without a referent. Such usage is commonplace. However, it is not ideal and writing clarity is universally improved by adding a referent noun or revising the construction of the sentence so that it does not have the demonstrative pronoun "this" without a referent noun. To improve the manuscript readability in the final manuscript submitted for production, please go through usages of "this" throughout the manuscript and when there is no referent noun, add noun referents or revise accordingly.

---

## [Editor Report · Acceptance letter]

7 Feb 2024

PCOMPBIOL-D-23-01558R1 

Accounting for Isoform Expression Increases Power to Identify Genetic Regulation of Gene Expression

Dear Dr LaPierre,

I am pleased to inform you that your manuscript has been formally accepted for publication in PLOS Computational Biology. Your manuscript is now with our production department and you will be notified of the publication date in due course.

With kind regards,

Anita Estes
